

# Nonequilibrium steady-state dynamics
# of Markov processes on graphs

Stefano Crotti[1*], Thomas Barthel[2,3†] and Alfredo Braunstein[1,4‡]

**1** Politecnico di Torino, Corso Duca degli Abruzzi 24, 10124 Torino
**2** Department of Physics, Duke University, Durham, North Carolina 27708, USA
**3** National Quantum Laboratory, University of Maryland, College Park, MD 20742, USA
**4** Italian Institute for Genomic Medicine, IRCCS Candiolo,
SP-142, I-10060, Candiolo (TO), Italy

★ stefano.crotti@polito.it , † thomas.barthel@duke.edu , ‡ alfredo.braunstein@polito.it

## Abstract

We propose an analytic approach for the steady-state dynamics of Markov processes on locally tree-like graphs. It is based on time-translation invariant probability distributions for edge trajectories, which we encode in terms of infinite matrix products. For homogeneous ensembles on regular graphs, the distribution is parametrized by a single $d \times d \times r^2$ tensor, where $r$ is the number of states per variable, and $d$ is the matrix-product bond dimension. While the method becomes exact in the large-$d$ limit, it typically provides highly accurate results even for small bond dimensions $d$. The $d^2 r^2$ parameters are determined by solving a fixed point equation, for which we provide an efficient belief-propagation procedure. We apply this approach to a variety of models, including Ising-Glauber dynamics with symmetric and asymmetric couplings, as well as the SIS model. Even for small $d$, the results are compatible with Monte Carlo estimates and accurately reproduce known exact solutions. The method provides access to precise temporal correlations, which, in some regimes, would be virtually impossible to estimate by sampling.

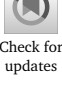

# 1  Introduction

Given a stationary Markov process described by the trajectory probability

$$p^T\left(\boldsymbol{x}^0,\ldots,\boldsymbol{x}^T\right)=\varphi(\boldsymbol{x}^0)\prod_{t=0}^{T-1}w\left(\boldsymbol{x}^{t+1}|\boldsymbol{x}^t\right), \tag{1}$$

characterizing its steady-state distribution and steady-state dynamics is a crucial task with a myriad of applications. Here,

$$\boldsymbol{x}^t=(x_1^t,\ldots,x_N^t), \quad \text{with} \quad x_i^t\in\{1,\ldots,r\}, \tag{2}$$

denotes the system state at time $t$, $w\left(\boldsymbol{x}'|\boldsymbol{x}\right)$ is the stochastic transition matrix, and $\varphi$ is a probability measure at $t=0$.

    For large systems, a direct manipulation of the $r^N\times r^N$ transition matrix $w$ to find its dominant eigenvector is computationally infeasible. While a direct Markov-chain Monte Carlo (MCMC) simulation might appear straightforward, unfortunately, it can be hampered by several factors, including the difficulty of estimating expectation values which are small due to cancellation effects and the very slow convergence to the steady state in many relevant scenarios. Consequently, analytical approximation schemes are often preferred. In the following, we focus on locally tree-like systems, such as random regular graphs, Erdős-Rényi graphs, and Gilbert graphs. If the Markov model (1) satisfies detailed balance $w(\boldsymbol{x}'|\boldsymbol{x})\varrho(\boldsymbol{x})=w(\boldsymbol{x}|\boldsymbol{x}')\varrho(\boldsymbol{x}')$ with respect to an equilibrium measure $\varrho$, the problem becomes more tractable. In this case, describing the steady state reduces to studying the equilibrium distribution, for which several analytic approximation methods have been developed in past years. These include the cavity and replica methods [1, 2] and their single-instance counterpart, belief propagation (BP) as well as its generalizations [3, 4]. For systems lacking detailed balance, such as non-symmetric Glauber dynamics or the susceptible-infectious-susceptible (SIS) model, one alternative to MCMC is provided by mean-field approximations. These typically yield a simplified set of dynamical equations for a reduced set of local variables. However, mean-field approximations are usually inaccurate when the interaction graph is sparse or when the process is state recurrent, i.e., when particles can return to previously visited states. The field is in active development, and several corrections have been proposed in recent years [5–9].

Recently, a new approximation method called matrix-product belief propagation was introduced to characterize dynamical transients [10–12]. This approach builds on the dynamical cavity method [6,13,14] and approximates the underlying edge messages – conditional probabilities for trajectories on neighboring vertices – in matrix-product form. These matrix-product edge messages are constructed iteratively, adding one tensor per time step. For a fixed bond dimension $d$, the number of variables grows linearly in the maximum time $T$ and the total computation cost is quadratic in $T$ due to truncations in each step. While matrix-product belief propagation is very effective for studying transient dynamics, it is inefficient for the investigation of steady-state dynamics.

In this work, we resolve this challenge by taking the infinite-time limit and introducing an approximation for time-translation invariant edge messages in terms of infinite matrix-product (iMP) distributions. This approach makes it possible to access the nonequilibrium steady-state dynamics and the continuous-time limit directly. Each iMP edge message is parametrized by a single $d \times d \times r^2$ tensor and is determined by a fixed point equation.

## 2 Infinite matrix-product edge messages

Let us consider a Markov process of the form (1) for a system living on a graph $G = (V, E)$ with vertices $V = \{1, 2, \ldots, N\}$ and edges $E \subset V \times V$, where the transition matrix takes the local form

$$w\left(\boldsymbol{x}^{t+1}|\boldsymbol{x}^t\right) = \prod_{i=1}^{N} w_i\left(x_i^{t+1}|\boldsymbol{x}_{\partial i}^t, x_i^t\right), \tag{3}$$

with $\partial i$ denoting the nearest neighbors of vertex $i$ and $\boldsymbol{x}_{\partial i}^t$ their state at time $t$, i.e., transitions happen in parallel for all variables with the update depending only on the state of neighbors. The continuous-time scenario will be addressed later. In the framework of belief propagation, the joint distribution (1) is reorganized in terms of single-variable trajectories $\overline{x}_i = (x_i^0, \ldots, x_i^T)$ and factors $f_i(\overline{x}_i, \overline{\boldsymbol{x}}_{\partial i}) = \varphi(x_i^0) \prod_{t=0}^{T-1} w_i\left(x_i^{t+1}|\boldsymbol{x}_{\partial i}^t, x_i^t\right)$. Then, assuming the graph $G$ to be tree-like gives the self-consistent set of belief-propagation equations [6, 12–14] for the edge messages

$$m_{i \to j}(\overline{x}_i, \overline{x}_j) = \sum_{\overline{\boldsymbol{x}}_{\partial i \setminus j}} f_i(\overline{x}_i, \overline{\boldsymbol{x}}_{\partial i}) \prod_{k \in \partial i \setminus j} m_{k \to i}(\overline{x}_k, \overline{x}_i). \tag{4}$$

Message $m_{i \to j}(\overline{x}_i, \overline{x}_j)$ is the probability for trajectory $\overline{x}_i$ on vertex $i$ given trajectory $\overline{x}_j$ on neighbor $j$ for the "cavity" system where all terms in factor $f_j$ are removed from the dynamical distribution (1) [11,13]. As all $\overline{x}_j$ have equal probability when $f_j$ is removed, $m_{i \to j}$ is also the *joint* probability of $\overline{x}_i$ and $\overline{x}_j$ in the cavity system. For state-recurrent dynamics as in the SIS model, belief propagation (4) suffers from an exponential growth of the number of trajectories with time $T$ and the resulting exponential computational complexity.

To overcome this obstacle and access nonequilibrium steady states, we take the limit $T \to \infty$ such that the initial state becomes irrelevant. Consequently, the edge messages become time-translation invariant, and we make the iMP ansatz

$$m_A(\overline{x}_i, \overline{x}_j) := \ldots A(x_i^t, x_j^t) A(x_i^{t+1}, x_j^{t+1}) \ldots, \tag{5}$$

characterized by a single $d \times d \times r \times r$ tensor $A_{a,b,x,y}$, and $A(x, y)$ is interpreted as a $d \times d$ matrix with tensor/matrix elements $[A(x, y)]_{a,b} = A_{a,b,x,y}$ such that Eq. (5) is an infinite product of matrices. The *bond dimension $d$* controls the computation costs and accuracy of the ansatz. This construction is analog to the finite-$T$ matrix-product edge messages from Refs. [10–12] and uniform infinite matrix product states used to encode spatial correlations of quantum many-body states [15–22]. For heterogeneous systems, one should work with edge-dependent

tensors $A = A_{i \to j}$. The boundary conditions in the matrix product (5) which map it to a scalar are generally irrelevant. Such technical details and an argument on the soundness of the iMP hypothesis are provided in Appendices C-G.

## 3 Belief propagation and truncations

We want to solve the belief propagation equation (4) in a fixed-point iteration. Inserting the iMP ansatz (5), we obtain the updated edge messages in the modified matrix product form

$$\tilde{m}_B(\overline{x}_i, \overline{x}_j) = \dots B(x_i^t, x_i^{t-1}, x_j^{t-1}) B(x_i^{t+1}, x_i^t, x_j^t) \dots, \tag{6}$$

with [10–12]

$$B_{i \to j}(x_i', x_i, x_j) = \sum_{\boldsymbol{x}_{\partial i \setminus j}} w_i(x_i' | \boldsymbol{x}_{\partial i}, x_i) \bigotimes_{k \in \partial i \setminus j} A_{k \to i}(x_k, x_i), \tag{7}$$

where $x_i$ and $x_i'$ are variables for times $t$ and $t+1$, respectively and $\bigotimes$ denotes the Kronecker product. We can recast the updated edge message (6) into the form (5) by an exact SVD or QR decomposition,

$$B(x_i^{t+1}, x_i^t, x_j^t) = Q(x_i^t, x_j^t) R(x_i^{t+1}), \tag{8}$$

to obtain the updated iMP tensor

$$\tilde{A}(x_i^t, x_j^t) := R(x_i^t) Q(x_i^t, x_j^t), \tag{9}$$

with bond dimension $\tilde{d} = r d^{|\partial i| - 1}$.

Iterating equations (7)-(9) naively would result in an exponential explosion of the bond dimension. It is therefore necessary to perform a truncation that approximates the target iMP message $m_{\tilde{A}}$ by one with a smaller bond dimension. Fortunately, excellent solutions have been developed [23, 24]. Here, we employ the variational uniform matrix product state (VUMPS) algorithm [22, 24, 25], maximizing the fidelity per time step with respect to an iMP message $m_A$ with the original bond dimension $d$ such that the new $d \times d \times r^2$ tensor $A$ is given by

$$\underset{A}{\arg\max} \lim_{T \to \infty} \left( \frac{|\langle m_A^T | m_{\tilde{A}}^T \rangle|}{\|m_A^T\| \|m_{\tilde{A}}^T\|} \right)^{1/T}, \tag{10}$$

where $m_A^T(\overline{x}_i, \overline{x}_j) := \text{Tr}[A(x_i^1, x_j^1) \dots A(x_i^T, x_j^T)]$ are $T$-cyclic matrix-product edge messages, we defined the inner product $\langle m | \tilde{m} \rangle := \sum_{\overline{x}_i, \overline{x}_j} m^*(\overline{x}_i, \overline{x}_j) \tilde{m}(\overline{x}_i, \overline{x}_j)$, and 2-norm $\|m\| := \langle m | m \rangle^{1/2}$. The quantity $|\langle m_A^T | m_{\tilde{A}}^T \rangle|^{1/T}$ converges for $T \to \infty$ to the maximum-magnitude eigenvalue of the $\tilde{d} d \times \tilde{d} d$ matrix $\sum_{x_i, x_j = 1}^r \tilde{A}(x_i, x_j) \otimes A^*(x_i, x_j)$. The VUMPS method iteratively solves a set of equations guaranteeing stationarity of the fidelity per time step by repeatedly solving the principal eigenvalue problems of $\sum_{x_i, x_j = 1}^r \tilde{A}(x_i, x_j) \otimes A^*(x_i, x_j)$ and $\sum_{x_i, x_j = 1}^r A(x_i, x_j) \otimes A^*(x_i, x_j)$. See [22, 24, 25] for more details. For consistency with the existing literature we use the complex conjugation symbol $^*$ even if the matrices considered in this work are real-valued. Although this was not explored here, note that complex-valued matrices are in principle more expressive and could offer some advantage to parametrize even real functions.

The computation costs can be reduced further from an exponential to a linear scaling in the vertex degree $|\partial i|$. This is achieved by contracting the edge messages $m_{k \to i}$ with $k \in \partial i \setminus j$ in Eq. (4) in sequence and truncating the matrix product after each contraction [12].

The *eternal dynamic cavity* (EDC) equations (7)-(10) are iterated until convergence. The fixed point provides the distribution of edge trajectories (beliefs)

$b_{i,j}(\overline{x}_i, \overline{x}_j) = m_{A_{i \to j}}(\overline{x}_i, \overline{x}_j) m_{A_{j \to i}}(\overline{x}_j, \overline{x}_i)$ which are further marginalized to compute equilibrium observables or $n$-point temporal correlations as discussed in Appendix E. As is customary for cavity approximations, the method can be used to work directly in the limit of infinitely sized graphs via single-edge updates for regular graphs or a population dynamics approach.

## 4  Results

We first apply the algorithm to parallel Glauber dynamics of classical spin variables $x_i \equiv \sigma_i \in \{\pm 1\}$, governed by transitions

$$w_i\left(\sigma_i^{t+1}|\boldsymbol{\sigma}_{\partial i}^t\right) \propto e^{\beta \sigma_i^{t+1}\left(\sum_{j \in \partial i} J_{ij}\sigma_j^t + h_i\right)}, \tag{11}$$

with a uniform field $h_i = h$ and symmetric couplings $J_{ij} = J_{ji} = J$ on an infinite random regular graph of vertex degree 3. Due to the symmetry, the dynamics converges to (a marginal of) the equilibrium state of a related Ising model [12,26,27]. Figure 1 compares the average EDC magnetization and nearest-neighbor correlations with the equilibrium values that can be obtained via the standard equilibrium cavity method.[1] As expected, increasing the bond dimension results in convergence to the exact solution.

For the same system, Figure 2 shows the steady-state autocovariance

$$c_i(\Delta t) = \langle \sigma_i^t \sigma_i^{t+\Delta t} \rangle - \langle \sigma_i^t \rangle \langle \sigma_i^{t+\Delta t} \rangle, \tag{12}$$

at distances up to $\Delta t = 40$ epochs of the dynamics. We compare with Monte Carlo estimates on increasingly larger random graphs. The Monte Carlo accuracy degrades as the autocovariance decays exponentially in $\Delta t$ and is quickly overwhelmed by the sampling error.

Next, we turn to dynamics with nonequilibrium steady states. First, consider the Glauber dynamics (11) with non-reciprocal interactions $J_{ij} \neq J_{ji}$. As a simple system with this feature, we analyze an infinite regular bipartite graph with vertices $V = A \cup B$ and edges $E \subset A \times B$. The non-reciprocal coupling strengths $J_{ij}$ are $J_{A \to B}$ if $i \in A$ and $j \in B$, $J_{B \to A}$ if $i \in B$ and $j \in A$, and zero otherwise. The asymmetry is increased further by choosing different vertex degrees $z_A = 3$ and $z_B = 4$ for $A$ and $B$ vertices. Figure 3 shows the average magnetization for nodes in both blocks of the bipartition as well as the global average. Good agreement is observed with Monte Carlo simulations on a large random graph and sufficiently large times.

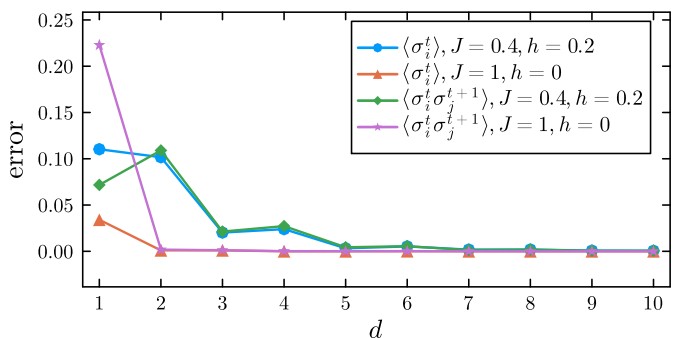

Figure 1: Errors for the EDC magnetization and nearest-neighbor correlations in the symmetric Glauber dynamics (11) on the infinite random 3-regular graph in the paramagnetic regime ($J = 0.4$, $h = 0.2$) and ferromagnetic regime ($J = 1$, $h = 0$). With increasing bond dimension $d$, the results converge to the equilibrium observables of the underlying Ising model obtained via the standard equilibrium cavity method.

---

[1] For the parallel dynamics, the correct quantity to compare with equilibrium correlations is $\langle \sigma_i^t \sigma_i^{t+1} \rangle$ instead of $\langle \sigma_i^t \sigma_j^t \rangle$ See S1 in the Supporting Information of ref. [12].

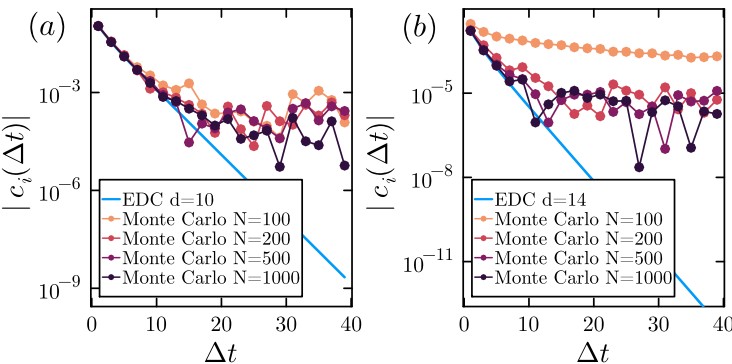

Figure 2: Equilibrium autocovariance (12) at time distance $\Delta t$ in Glauber dynamics (11) on an infinite random regular graph of degree 3. (a) Paramagnetic regime with $J = 0.4$, and $h = 0.2$. (b) Ferromagnetic regime with $J = 1$, $h = 0$. We compare EDC to Monte Carlo data with $10^4$ samples, time horizon $T_{\mathrm{MC}} = 101$, and finite graph sizes $N$. For Monte Carlo, the absolute value $|c_i(\Delta t)|$ is shown, because the sampling error causes estimated autocovariances to fluctuate below zero; this is not the case for the EDC solution. Monte Carlo error bars are very large for the larger $\Delta t$ and are omitted for clarity.

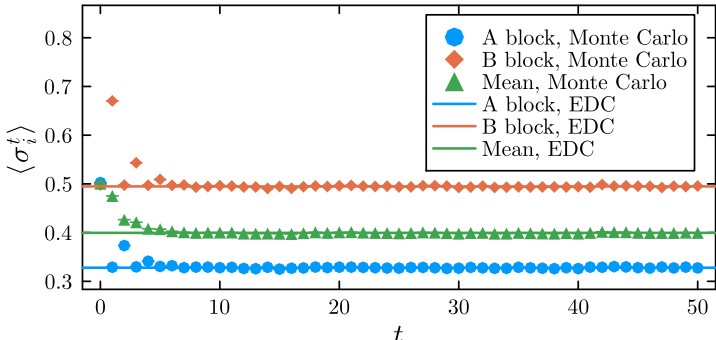

Figure 3: Parallel Glauber dynamics (11) on an infinite regular bipartite graph $G = (V = A \cup B, E)$ with non-reciprocal couplings $J_{B \to A} = 0.1$, $J_{A \to B} = 0.5$, external field $h = 0.2$, and vertex degrees $z_A = 3$, $z_B = 4$. Points correspond to the transient of a Monte Carlo simulation on a random graph with $N_A = 1200$ and $N_B = 900$ vertices. Error bars are smaller than symbols sizes. Lines show EDC results for $d = 5$.

Another example where an analytical expression for the nonequilibrium steady state is not known is the SIS model of epidemic spreading. The Markov rule for the time-discretized version of this model reads [28, 29]

$$w_i\left(x_i^{t+1} = S \mid \boldsymbol{x}_{\partial i}^t, x_i^t\right) = \rho\, \delta_{x_i^t, I} + \delta_{x_i^t, S} \prod_{j \in \partial i} \left(1 - \lambda\, \delta_{x_j^t, I}\right), \tag{13}$$

with variables $x_i \in \{S, I\}$ and the Kronecker delta $\delta$. For an infinite degree-3 random regular graph, a fixed recovery probability $\rho = 0.1$ and several values of the transmission probability $\lambda$, we compute the EDC probability for a node to be infectious in the steady state and also show deviations with respect to an extensive Monte Carlo simulation in Fig. 4. The performance is compared further with the discretized version of three mean-field approaches [12]: recurrent dynamic message passing (rDMP) [7], individual-based mean field (IBMF) [5] and the cavity master equation (CME) [8]. Our method achieves the best precision across the whole range of transmission probability, including $\lambda/\rho \approx 0.55$ which is close to a dynamical transition above which a sustained epidemic is the stable steady state. See Appendix A for more details.

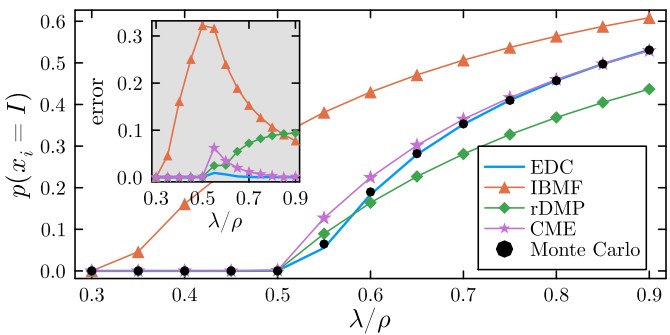

Figure 4: Probability of a vertex being infectious in the steady state of the SIS model on an infinite degree-3 random regular graph with recovery probability $\rho = 0.1$ and varying transmission probability $\lambda$. Here, the EDC solution with bond dimension $d = 20$ is compared to Monte Carlo and three mean-field approaches (see text). Inset: absolute error $|p(x_i = I) - p_{\mathrm{MC}}(x_i = I)|$ with respect to a Monte Carlo simulation. The Monte Carlo and mean-field methods are applied for a finite graph of size $N = 5000$, and for finite time horizons $T_{\mathrm{MC}} = 4000$ and $T_{\mathrm{MF}} = 10000$, respectively.

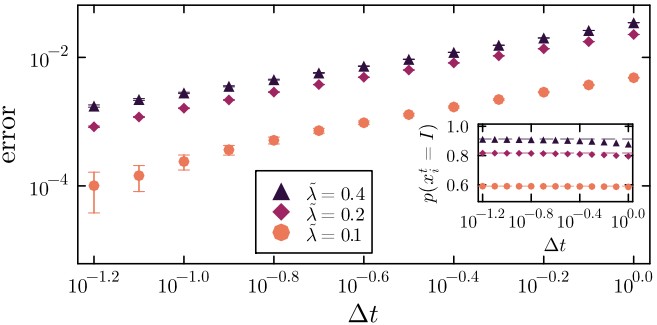

Figure 5: Continuous-time SIS model with recovery rate $\tilde{\rho} = 0.1$ and transmission rates $\tilde{\lambda} = 0.1, 0.2, 0.4$ on an infinite degree-3 random regular graph. We compare the accuracy of the EDC solution for bond dimension $d = 6$ and varying time discretization $\Delta t$ to a Gillespie Monte-Carlo simulation with graph size $N = 2000$ and time horizon $T_{\mathrm{MC}} = 10^7$. The main panel shows the deviations $|p(x_i = I) - p_{\mathrm{MC}}(x_i = I)|$ for single-time marginals. The inset shows the probabilities, where horizontal dashed lines indicate the Monte Carlo values.

## 5 Continuous-time and asynchronous dynamics

Finally, we show how the EDC method (7)-(10) can be applied to both continuous-time and asynchronous dynamics. Regarding the former, recall that SIS dynamics with transition rates $\tilde{\lambda}$ and $\tilde{\rho}$ on a continuous time interval $[0, T]$ can be defined as the $\Delta t \to 0$ limit of a discrete-time dynamics with $T/\Delta t$ epochs and transition probabilities $\lambda = \Delta t\, \tilde{\lambda}$ and $\rho = \Delta t\, \tilde{\rho}$. In the small-$\Delta t$ limit, discrete-time Monte-Carlo simulations become extremely expensive such that, usually, continuous-time alternatives like the Gillespie Monte Carlo method [30] are employed instead. The finite-$T$ matrix-product belief propagation [10–12] would also suffer from this drawback, as the computation cost scales quadratically in the number of epochs $T/\Delta t$. Instead, we find that the EDC method, based on a single tensor $A$, with small $\Delta t$, can perfectly reproduce the continuous time steady-state dynamics without increasing the computation costs. Figure 5 compares the resulting steady-state probabilities of being infectious. Note that it is difficult to evaluate accuracies at very small $\Delta t$ due to the Monte-Carlo sampling error.

It should be noted that taking the $\Delta t \to 0$ limit of the equations is in principle possible and could be preferable. However, the limit presents some non-trivial technical challenges related to parametrizing probability distributions of continuous variables through continuous matrix products (see e.g. [31]) and their truncations. This will be addressed in future work.

Similar results can be obtained for asynchronous dynamics. Indeed, when replacing $w(x_i^{t+1}|x_{\partial i}^t, x_i^t)$ in (3) by $w'(x_i^{t+1}|\boldsymbol{x}_{\partial i}^t, x_i^t) = \rho \delta_{x_i^t, x_i^{t+1}} + (1-\rho)w(x_i^{t+1}|\boldsymbol{x}_{\partial i}^t, x_i^t)$, the steady state of the parallel dynamics converges to the one of the asynchronous dynamics in the limit $\rho \to 1$.

# 6 Discussion

We have demonstrated how steady-state dynamics on locally tree-like graphs can be studied efficiently through the probability distribution of infinitely long trajectories of the system. This distribution can be analyzed by solving dynamic belief propagation equations (4) with an infinite matrix-product ansatz (5) for the edge messages. In general, the computational complexity for recurrent-state dynamics with nonequilibrium steady states scale exponentially with both system size $N$ and time horizon $T$. The EDC method overcomes the exponential $N$ dependence and directly operates in the $T \to \infty$ limit, with computational costs depending instead on temporal fluctuations through the required bond dimension $d$. For regular graphs and homogeneous transition rules, one can work with a single edge message, characterized by a single $d \times d \times r^2$ tensor. This enables analytical investigations and, due to a much more favorable error scaling compared to Markov-chain Monte Carlo, the method makes it possible to efficiently analyze correlation times and dynamic scaling exponents. Important applications concern, for example, the endemic phases of infectious diseases, kinetically constrained systems used to model glassy materials [32], exclusion processes in biology [33], opinion dynamics [34], linear threshold and cascade models [13, 35], as well as nonequilibrium solvers for optimization problems [36]. Code for the algorithm is available at [37–39]. For heterogeneous and disordered systems, it is straightforward to combine the approach with population dynamics [3], working with one matrix-product edge message (5) for each class of equivalent edges.

When taking the $T \to \infty$ limit, some care may be necessary to ensure that the dynamics under consideration converges to a desired unique stationary state. Appendix A discusses the issue for SIS dynamics. The computational cost of the algorithm and for the evaluation of observables, while scaling favorably with vertex degree and time window lengths, generally grows as $O(d^6)$ [11, 12]. Based on experience with matrix-product methods for quantum many-body groundstate problems, we expect that the EDC bond dimensions have to grow according to a power law $d \sim |g - g_c|^{-\eta}$ when approaching a dynamic phase transition at a critical model parameter $g_c$. Appendix B provides corresponding data for required bond dimensions near the critical points in Glauber-Ising and SIS dynamics. At small $d$, one may encounter matrix-product transfer matrices with degenerate principal eigenvalues. A simple way to avoid corresponding complications is to change $d$; see Appendix D for details.

Note that belief propagation has very recently also emerged as a useful tool for gauge fixing and the evaluation of expectation values for tensor networks that describe quantum ground states or classical thermal states of many-body systems [40–47]. The algorithm pursued here and in Refs. [10–12] applied to a $D$-dimensional graph, can be used as a belief propagation for $D + 1$ dimensional tensor networks, e.g., expectation values of projected entangled-pair states (PEPS) or partition sums of classical systems with translation invariance in (at least) one direction.

# Acknowledgments

SC thanks Lander Burgelman for his help with the usage of VUMPS software.

**Funding information** This study was carried out within the FAIR - Future Artificial Intelligence Research project and received funding from the European Union Next-GenerationEU (Piano Nazionale di Ripresa e Resilienza (PNRR)–Missione 4 Componente 2, Investimento 1.3–D.D. 1555 11/10/2022, PE00000013). This manuscript reflects only the authors' views and opinions, neither the European Union nor the European Commission can be considered responsible for them. TB gratefully acknowledges support by the Duke Population Research Center (DPRC), the U.S. NICHD grant P2C-HD0065563, and the U.S. NSF grant DMS-2344576. We acknowledge support from the European REA, Marie Skłodowska-Curie Actions, grant agreement no. 101131463 (SIMBAD).

# A   Stationary states of SIS dynamics

SIS dynamics on finite graphs only have one "true" stationary state – the absorbing state, where all individuals are susceptible. Once in this absorbing state, the system cannot escape it. Furthermore, starting from any other configuration, the system has non-zero probability of eventually transitioning to the absorbing state, making it the unique stationary state according to the Perron-Frobenius theorem [48]. There exist, however, other quasi-stationary states, corresponding to an endemic regime of the epidemic [49]. These are states where one observes a finite fraction of infectious individuals at long times. It is bound to eventually die out but, on large graphs, the epidemic is sustained for long enough times to be worth studying. This situation closely resembles the phenomenon of endemic diseases observed in nature. For this reasons, the interest is often directed to quasi-stationary endemic states rather than the trivial absorbing state. To this purpose finite-size methods are endowed with corrections to discard the absorbing state [8,49]. The situation is somewhat different for infinite-size graphs as, in the proper regime, true endemic states can exist [50]. In particular, this implies that the eternal dynamic cavity (EDC) equations (7)-(10) for infinite regular graphs also have both types of fixed points.

In both the finite and infinite cases, one would like to divert the dynamics away from the absorbing state to study the more interesting endemic one. The technique employed here is to add a small auto-infection probability $\alpha$ which allows spontaneous transitions away from the all-susceptible state. The modified Markov transition reads

$$w_i\left(x_i^{t+1}=S|\boldsymbol{x}_{\partial i}^t, x_i^t\right) = \rho\,\delta_{x_i^t, I} + (1-\alpha)\delta_{x_i^t, S}\prod_{j\in\partial i}\left(1-\lambda\,\delta_{x_j^t, I}\right), \tag{A.1}$$

with the probability for $x_i^{t+1} = I$ following from the normalization $\sum_{x_i'} w_i(x_i'|\boldsymbol{x}_{\partial i}, x_i^t) = 1$. We observe that it can be beneficial to start the EDC method with a small auto-infection, say $\alpha = 0.1$, and to then gradually lower it to zero as the fixed point is approached.

# B   Bond dimension

Accurately capturing the dynamics of a system near a phase transition can present challenges. We think that this is related to time correlations becoming long-ranged. In analogy to what happens in quantum systems, where long-range spatial correlations require larger bond dimensions [51], we argue that a similar situation arises in the EDC method with respect to

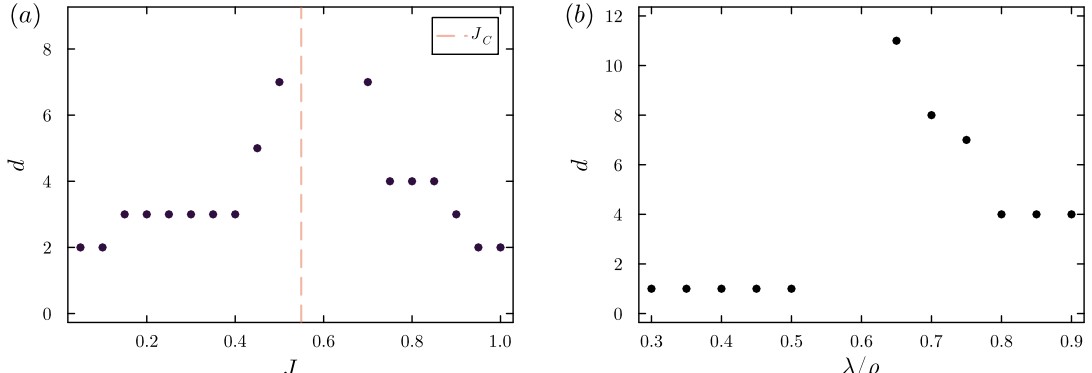

Figure 6: Analysis of the bond dimension $d$ required to achieve a specified precision for EDC single-time, single-site observables. (a) Glauber dynamics (11) on an infinite degree-3 random regular graph with zero external field $h = 0$, where the coupling strength $J_{ij} \equiv J$ is varied. The shown value of $d$ is the smallest for which both, the error of the single-site magnetization $\langle \sigma_i^t \rangle$ and the pair correlation $\langle \sigma_i^t \sigma_j^{t+1} \rangle$ are below $10^{-3}$. (b) SIS model (A.1) on an infinite degree-3 random regular graph with recovery probability $\rho = 0.1$ and varying transmission probability $\lambda$. The value of $d$ is the smallest for which the error on the single-site probability $p(x_i^t = I)$ falls below $10^{-3}$.

temporal correlations. Figure 6 illustrates the bond dimension $d$ in the iMP ansatz (5) required to achieve a specified precision of the EDC single-time marginals.

For symmetric Glauber dynamics (11) with $J_{ij} \equiv J$ on a degree-$k$ random-regular graph, the underlying zero-field Ising model undergoes a ferromagnetic transition at the critical coupling strength $J_c = (\log \frac{k}{k-2})/2$, which is approximately $J_c \approx 0.5493$ for $k = 3$. Figure 6a demonstrates the increase in bond dimension needed to achieve an accuracy within $10^{-3}$ of the equilibrium magnetization when approaching $J_c$. This behavior is consistent with the finite-$T$ results presented in Ref. [11, Section 7].

A similar scenario arises when considering the SIS model (A.1) on a degree-3 random-regular graph. There exists a critical transmission probability $\lambda_c$ with $0.5 < \lambda_c/\rho < 0.6$. Below this threshold, the only stable state is one where all individuals are susceptible. On an infinite graph, the epidemic persists indefinitely for $\lambda > \lambda_c$. Figure 6b illustrates that obtaining accurate estimates is more challenging in this region and the required bond dimension increases when approaching $\lambda_c$ from above.

## C  Infinite matrix-product ansatz for steady-state edge messages

We provide here an argument as to why the infinite matrix-product (iMP) ansatz (5) for the edge messages is appropriate and in what sense it becomes exact in the limit of large bond dimension $d$. Within the constraints of the cavity method (approximate treatment of loops in the graph), the goal is to efficiently capture the trajectory distribution $p^T$ in Eq. (1) in the limit $T \to \infty$. We will first see that one can construct a modified time-cyclic distribution $q^T$ which is equivalent to $p^T$ in the sense that all finite-time statistics of both are equal in the limit

$T \to \infty$. Specifically, let

$$q^T(\underbrace{x^0,\dots,x^T}_{=:\overline{x}}) := \frac{1}{Z_T} w(x^0|x^T) \prod_{t=0}^{T-1} w(x^{t+1}|x^t), \tag{C.1}$$

where, defining the $r^N \times r^N$ transition matrix $W$ as $W_{x',x} := w(x'|x)$, $Z_T = \mathrm{Tr}(W^T)$ normalizes the nonnegative $q^T$ such that it is a proper probability. If the transition matrix (3) is irreducible, the time-cyclic distribution (C.1) recovers the original dynamics in the infinite-time limit as the marginals for the variables $(x^{T-t+1},\dots,x^T)$ of any finite time interval agree,

$$\lim_{T\to\infty} \sum_{x^0,\dots,x^{T-t}} q^T(\overline{x}) = \lim_{T\to\infty} \sum_{x^0,\dots,x^{T-t}} p^T(\overline{x}). \tag{C.2}$$

Equation (C.2) can be shown as follows. With the canonical vector basis $\{e_x\}$, and vector $\varphi$ denoting the $t=0$ distribution from Eq. (1), we have

$$\sum_{x^0,\dots,x^{T-t}} p^T(\overline{x}) = \sum_y w(x^T|x^{T-1}) w(x^{T-1}|x^{T-2})\dots w(x^{T-t+1}|y)\, e_y^{\mathsf{T}} W^{T-t} \varphi, \quad \text{and} \quad \tag{C.3a}$$

$$\sum_{x^0,\dots,x^{T-t}} q^T(\overline{x}) = \frac{1}{Z_T} \sum_y w(x^T|x^{T-1}) w(x^{T-1}|x^{T-2})\dots w(x^{T-t+1}|y)\, e_y^{\mathsf{T}} W^{T-t} e_{x^T}. \tag{C.3b}$$

According to the Perron-Frobenius theorem [48], the transition matrix $W$ has a unique stationary measure $\pi$. Hence,

$$\lim_{T\to\infty} W^{T-t}\varphi = \lim_{T\to\infty} W^{T-t} e_{x^T} = \pi, \quad \text{and} \tag{C.4a}$$

$$\lim_{T\to\infty} Z_T = \lim_{T\to\infty} \mathrm{Tr}(W^T) = \mathrm{Tr}(\pi e^{\mathsf{T}}) = \sum_x \pi(x) = 1, \tag{C.4b}$$

with the one-vector $e = (1,1,\dots,1)$. So, the two expressions (C.3) agree for $T \to \infty$, i.e., we find Eq. (C.2).

The advantage of the distribution $q^T$ is that it is manifestly time-translation invariant, i.e., $q^T(x^0,\dots,x^T) = q^T(x^t,\dots,x^{T+t})$ for every $t = 0,\dots,T$, where $x^{t+T} \equiv x^t$. Hence, applying belief propagation (4) for the modified dynamics (C.1), the resulting edge messages are also time-cyclic invariant,

$$m_{i\to j}^T\left((x_i^0,\dots,x_i^T),(x_j^0,\dots,x_j^T)\right) = m_{i\to j}^T\left((x_i^t,\dots,x_i^{t+T}),(x_j^t,\dots,x_j^{t+T})\right). \tag{C.5}$$

Now, as shown in Ref. [52, Theorem 3] and detailed in Appx. G, every such cyclic edge message has an exact uniform matrix-product representation

$$m_A^T(\overline{x}_i,\overline{x}_j) = \mathrm{Tr}\left[A\left(x_i^0,x_j^0\right)\dots A\left(x_i^T,x_j^T\right)\right], \quad \text{with} \quad A(x,x') \in \mathbb{R}^{d\times d}, \quad \text{and} \quad d \le 2r^{T+1}. \tag{C.6}$$

The final step to arrive at the iMP edge message (5) is to take the infinite-time limit. While we will keep the form (C.6), the bond dimension $d$ will generally diverge for exact matrix-product representations. Retaining a finite $d$ when $T \to \infty$ is generally an approximation. However, due to a decay of temporal correlations in the edge messages, the approximation error typically decays exponentially with increasing $d$.

## D  Definition of iMP distributions and boundary conditions

**The non-degenerate case.** We will give here a precise definition of iMP probability distributions as in the expression (5). Consider an iMP distribution (5) characterized by a tensor

$A \in \mathbb{R}^{d \times d \times k}$, which can be interpreted as a matrix-valued function $A : \{1, \ldots, k\} \to \mathbb{R}^{d \times d}$ with $k = d^2$, and we define the variable tuples $y^t := (x_i^t, x_j^t)$. Let us assume here that the principal eigenspace of the transfer matrix

$$F_A := \sum_{y=1}^{k} A(y), \tag{D.1}$$

has dimension 1. The expression

$$m_A(\overline{y}) := \ldots A(y^t) A(y^{t+1}) \ldots, \tag{D.2}$$

defines $m_A$ as a probability measure on infinite trajectories $\overline{y} = (\ldots, y^t, y^{t+1}, \ldots) \in \{1, \ldots, k\}^{\mathbb{Z}}$. The measure cannot be defined on single trajectories. As the space of trajectories is uncountable, generally every single trajectory has probability zero. In an analogous way to the definition of an infinite-product measure space, we define the measure on the $\Sigma$−algebra generated by hyper-cubes $U_{\tilde{y}^t, \ldots, \tilde{y}^{t+\Delta t}} = \{\overline{y} \mid y^t = \tilde{y}^t, \ldots, y^{t+\Delta t} = \tilde{y}^{t+\Delta t}\}$. Let $\ell$ and $r$ denote the left and right principal eigenvectors of the transfer matrix $F_A$ such that $F_A r = \lambda r$ and $\ell^\intercal F_A = \lambda \ell^\intercal$. We define the measure of $U_{y^t, \ldots, y^{t+\Delta t}}$ as

$$m_A(y^t, \ldots, y^{t+\Delta t}) := \frac{1}{z} \ell^\intercal A(y^t) A(y^{t+1}) \ldots A(y^{t+\Delta t}) r, \quad \text{with} \quad z = \lambda^{\Delta t + 1} \ell^\intercal r, \tag{D.3}$$

where the non-degeneracy of $\lambda$ ensures that $z \neq 0$.

While the computation of finite-point marginals of an iMP distribution (D.2) is immediate thanks to Eq. (D.3), other observables need to be evaluated, first, with the finite $T$ representation [cf. also Eq. (C.6)]

$$m_A^T(y^0, \ldots, y^T) := \frac{1}{z_T} \text{Tr}[A(y^0) \ldots A(y^T)], \quad \text{with} \quad z_T = \text{Tr}(F_A^{T+1}), \tag{D.4}$$

and the $T \to \infty$ limit is to be taken afterward. In the large-$T$ limit, finite-time marginals of cyclic uniform matrix-product distributions (D.4) converge to (D.3). Indeed,

$$\sum_{y^{\Delta t+1}, \ldots, y^T} m_A^T(y^0, \ldots, y^T) = \frac{1}{z_T} \text{Tr}[A(y^0) \ldots A(y^{\Delta t}) F_A^{T-\Delta t-1}]$$

$$\xrightarrow{T \to \infty} \frac{1}{z} \text{Tr}[A(y^0) \ldots A(y^{\Delta t}) r \ell^\intercal] = \frac{1}{z} \ell^\intercal A(y^0) \ldots A(y^{\Delta t}) r. \tag{D.5}$$

**The degenerate case.** Transfer matrices like $F_A$ from Eq. (D.1) can in general have a degenerate dominant eigenvalue. Note that degenerate matrices are a set of measure zero, so an infinitesimal random perturbation brings it back to the non-degenerate case. In practice, a small change of the bond dimension $d$ is usually sufficient to resolve such degeneracies for EDC solutions. If one wishes to treat the degenerate case more systematically, one can replace (D.4) by

$$m_{A,Q}^T(y^0, \ldots, y^T) := \frac{1}{z_T} \text{Tr}[A(y^0) \ldots A(y^{\lfloor T/2 \rfloor}) Q A(y^{\lfloor T/2 \rfloor + 1}) \ldots A(y^T)], \tag{D.6}$$

with $z_T = \text{Tr}(Q F_A^{T+1})$ and a boundary matrix $Q \in \mathbb{R}^{d \times d}$, and define marginals through a limit analogous to Eq. (D.5). The latter generally depend on the choice of $Q$.

Note that, *in the non-degenerate case*, marginals of Eq. (D.6) still converge to Eq. (D.5) irrespective of the particular boundary term $Q$, provided that $\ell^\intercal Q r \neq 0$. A different $Q$ is to be chosen if this condition is not satisfied. Moreover, for the choice $Q = r \ell^\intercal$, one exactly recovers Eq. (D.4) even for finite $T$. In this work, we generally assume non-degeneracy for the dominant eigenvalues of the transfer matrices that occur in the evaluation of observables (cf. Appx. E) and the fidelity maximization (10). We simply increment $d$ when the EDC equations (7)-(10) do not converge, which is an expected consequence of degeneracies. This situation was observed only for very small $d$.

**Products of edge messages.** Two iMP probability distributions can be multiplied point-wise. Indeed, any finite marginal of the product $m_A^T(y^0, \ldots, y^T) m_B^T(y^0, \ldots, y^T)$ has a well-defined limit,

$$\frac{1}{z_T} \sum_{y^{\Delta t+1}, \ldots, y^T} \text{Tr}\big[A(y^0) \ldots A(y^T)\big] \text{Tr}\big[B(y^0) \ldots B(y^T)\big]$$

$$= \frac{1}{z_T} \sum_{y^{\Delta t+1}, \ldots, y^T} \text{Tr}\left[\prod_{s=0}^{T} A(y^s) \otimes B(y^s)\right] = \frac{1}{z_T} \text{Tr}\left[\prod_{s=0}^{\Delta t} A(y^s) \otimes B(y^s) F_{A\otimes B}^{T-\Delta t-1}\right]$$

$$\xrightarrow{T \to \infty} \frac{1}{z} \text{Tr}\left[\prod_{s=0}^{\Delta t} A(y^s) \otimes B(y^s) \boldsymbol{r}\boldsymbol{\ell}^\intercal\right] = \frac{1}{z} \boldsymbol{\ell}^\intercal \left(\prod_{s=0}^{\Delta t} A(y^s) \otimes B(y^s)\right) \boldsymbol{r},$$

where $\boldsymbol{\ell}$ and $\boldsymbol{r}$ are respectively the left and right dominant eigenvectors of $F_{A\otimes B}$ (provided that its dominant eigenvalue is non-degenerate), and $z_T = \text{Tr}(F_{A\otimes B}^{T+1})$. In short, we can consistently define $m_A m_B := m_{A\otimes B}$. As in Eq. (6), one typically uses the eloquent shorthand notation

$$\big[\ldots A(y^0) A(y^1) \ldots\big]\big[\ldots B(y^0) B(y^1) \ldots\big] = \ldots \big[A(y^0) \otimes B(y^0)\big]\big[A(y^1) \otimes B(y^1)\big] \ldots, \quad \text{(D.7)}$$

for products of iMP distributions.

Let us emphasize again that, while the iMP distributions define probability distributions on the space of infinite trajectories, expressions such as Eqs. (D.2) and (D.7) are formal in the sense that they do not denote the probability of a single trajectory. Indeed, the probability of any single infinite trajectory is generally zero. The real meaning of such expressions is that the marginals for any finite-time interval are given by Eq. (D.5). Observables should be evaluated from the finite-$T$ version (D.4) and by, then, taking the $T \to \infty$ limit of the result.

# E  Evaluation of observables

Given the two iMP messages (5) for edge $(i, j)$, the joint probability for trajectories $\overline{x}_i$ and $\overline{x}_j$ is [10–12]

$$b_{i,j}(\overline{x}_i, \overline{x}_j) = m_{A_{i\to j}}(\overline{x}_i, \overline{x}_j) m_{A_{j\to i}}(\overline{x}_j, \overline{x}_i)$$

$$= \ldots E(x_i^t, x_j^t) E(x_i^{t+1}, x_j^{t+1}) \ldots, \quad \text{(E.1a)}$$

$$\text{with} \quad E(x_i, x_j) := A_{i\to j}(x_i, x_j) \otimes A_{j\to i}(x_j, x_i). \quad \text{(E.1b)}$$

For any time interval $I_{\Delta t} = \{0, \ldots, \Delta t\}$, we can then compute the marginals in analogy to Eq. (D.5),

$$b_{i,j}\Big(\big(x_i^0, \ldots, x_i^{\Delta t}\big), \big(x_j^0, \ldots, x_j^{\Delta t}\big)\Big) := \sum_{\{x_i^t, x_j^t \mid t \notin I_{\Delta t}\}} b_{i,j}(\overline{x}_i, \overline{x}_j)$$

$$= \ldots FFF\, E\big(x_i^0, x_j^0\big) \ldots E\big(x_i^{\Delta t}, x_j^{\Delta t}\big) FFF \ldots \quad \text{(E.2)}$$

$$= \frac{1}{z} \boldsymbol{\ell}^\intercal E\big(x_i^0, x_j^0\big) \ldots E\big(x_i^{\Delta t}, x_j^{\Delta t}\big) \boldsymbol{r},$$

with the transfer matrix

$$F \equiv F_{A_{i\to j} \otimes A_{j\to i}} = \sum_{x, x'} E(x, x'), \quad \text{and} \quad z = \lambda^{\Delta t+1} \boldsymbol{\ell}^\intercal \boldsymbol{r}. \quad \text{(E.3)}$$

In Eq. (E.2), we have assumed that the transfer matrix has the non-degenerate dominant eigenvalue $\lambda$ with left and right eigenvectors $\boldsymbol{\ell}$ and $\boldsymbol{r}$. The belief (E.2) is the joint probability for

state sequences $\left(x_i^0, \ldots, x_i^{\Delta t}\right)$ and $\left(x_j^0, \ldots, x_j^{\Delta t}\right)$ on vertices $i$ and $j$ within the EDC approximation for the dynamics (1). From the beliefs, we can easily obtain time-local observables, time correlations, and edge-time correlations by further marginalization.

## F Exponential decay of correlations

The iMP representation (5) of edge messages $m_{A_{i\to j}}(\overline{x}_i, \overline{x}_j)$ obtained via EDC and the resulting iMP representation (E.1) for edge-trajectory probabilities $b_{i,j}(\overline{x}_i, \overline{x}_j)$ enables a direct estimation of correlation times. Let $\lambda_1, \lambda_2, \ldots$ denote the eigenvalues of the transfer matrix (E.3), ordered according to decreasing amplitude, and let $\boldsymbol{\ell}$ and $\boldsymbol{r}$ denote the dominant left and right eigenvectors. Using Eq. (E.2) and variable tuples $y^t := (x_i^t, x_j^t)$, the autocorrelation function for variables $y^0$ and $y^t$ evaluates to

$$
\begin{aligned}
b_{i,j}\left(y^0, y^t\right) - b_{i,j}\left(y^0\right) b_{i,j}\left(y^t\right) &= \frac{\boldsymbol{\ell}^\intercal E\left(y^0\right) F^{t-1} E\left(y^t\right) \boldsymbol{r}}{\lambda_1^{t+1} \boldsymbol{\ell}^\intercal \boldsymbol{r}} - \frac{\boldsymbol{\ell}^\intercal E\left(y^0\right) \boldsymbol{r}}{\lambda_1 \boldsymbol{\ell}^\intercal \boldsymbol{r}} \frac{\boldsymbol{\ell}^\intercal E\left(y^t\right) \boldsymbol{r}}{\lambda_1 \boldsymbol{\ell}^\intercal \boldsymbol{r}} \\
&= \frac{1}{\lambda_1^2 \boldsymbol{\ell}^\intercal \boldsymbol{r}} \boldsymbol{\ell}^\intercal E\left(y^0\right) \left[(F/\lambda_1)^{t-1} - \boldsymbol{r}\boldsymbol{\ell}^\intercal\right] E\left(y^t\right) \boldsymbol{r}.
\end{aligned}
\tag{F.1}
$$

Asymptotically, the matrix $(F/\lambda_1)^t - \boldsymbol{r}\boldsymbol{\ell}^\intercal$ decays exponentially as $|\lambda_2/\lambda_1|^t = e^{-t/\tau_2}$ with the correlation time $\tau_k := -1/\ln|\lambda_k/\lambda_1|$. In analogy to the spatial correlations in MPS as discussed in Refs. [51, 53], the iMP edge messages can nevertheless approximate a power-law decay of temporal correlations by a linear combination of exponential decays $\sim e^{-t/\tau_k}$.

Figure 7 shows results for characteristic correlation times evaluated in this way for the steady-state dynamics of the Glauber-Ising model (11) and the SIS model (A.1), demonstrating also that the bond dimension $d$ needs to be increased to capture correlations accurately upon approach of a critical point.

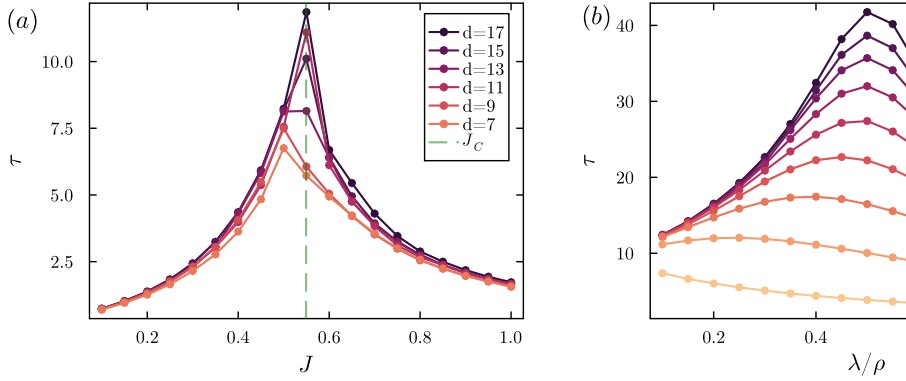

Figure 7: Characteristic correlation times obtained at different bond dimensions $d$. (a) Glauber dynamics (11) on an infinite degree-3 random regular graph with zero external field $h = 0$, where the coupling strength $J_{ij} \equiv J$ is varied. (b) SIS model (A.1) on an infinite degree-3 random regular graph with recovery probability $\rho = 0.1$ and varying transmission probability $\lambda$.

# G  Expressibility of cyclic matrix products

In Appx. C, we considered the time-cyclic edge messages in Eq. (C.5)

$$m^T\left(\left(x_i^0,\ldots,x_i^T\right),\left(x_j^0,\ldots,x_j^T\right)\right)=:m^T(y^0,\ldots,y^T),\quad\text{with}\quad y^t\equiv\left(x_i^0,x_j^0\right),\qquad\text{(G.1)}$$

which solve the belief propagation equations (4) for the modified dynamics (C.1). We showed that, in the limit $T\to\infty$, they yield the same marginals for finite time intervals as messages for the original dynamics (1); see Eq. (C.2). Following Ref. [52, Theorem 3], we want to show here that cyclic edge messages (G.1) have exact uniform matrix-product representations (C.6) with bond dimension $d\le 2r^{T+1}$. First, we can apply a sequence of exact SVDs or QR decompositions to sequentially split $m^T$ into a matrix product

$$
\begin{aligned}
m^T\left(y^0,\ldots,y^T\right)&\overset{\text{QR}}{=:}A_0\left(y^0\right)m^{(1,T)}\left(y^1,\ldots,y^T\right)\overset{\text{QR}}{=:}A_0\left(y^0\right)m^{(1,T-1)}\left(y^1,\ldots,y^{T-1}\right)A_T\left(y^T\right)\\
&\overset{\text{QR}}{=:}A_0\left(y^0\right)A_1(y^1)m^{(2,T-1)}(y_i^2,\ldots,y^{T-1})A_T\left(y_i^T\right)\overset{\text{QR}}{=:}\ldots\\
&\overset{\text{QR}}{=:}A_0\left(y^0\right)A_1\left(y^1\right)A_2\left(y^2\right)\ldots A_{T-1}\left(y^{T-1}\right)A_T\left(y_i^T\right).
\end{aligned}\qquad\text{(G.2)}
$$

For example, in the first equality, the decomposition is applied to the $r^2\times r^{2T}$ matrix $M_{y_0,(y_1,\ldots,y^T)}=m^T(y^0,\ldots,y^T)$. In the second equality, to $M_{(y^1,\ldots,y^{T-1},\alpha),(y^T,\beta)}=m^{(1,T)}(y^1,\ldots,y^T)_{\alpha,\beta}$, and so on. For $t=0,\ldots,\lfloor T/2\rfloor$, $A_t(y)$ are $r^{2t}\times r^{2t+2}$ matrices with $y\in\{1,\ldots,r^2\}$, and $A_{T-t}(y)$ are $r^{2t+2}\times r^{2t}$ matrices. So, the maximum bond dimension in the matrix product (G.2) is $r^{2\lceil T/2\rceil}$.

Due to the time-cyclic invariance (C.5) of the edge message $m^T$, we can now write it in the uniform matrix-product representation

$$
\begin{aligned}
m^T\left(y^0,\ldots,y^T\right)&=\frac{1}{T}\left(m^T\left(y^0,\ldots,y^T\right)+m^T\left(y^1,\ldots,y^T,y^0\right)+\cdots+m^T\left(y^T,y^0,\ldots,y^{T-1}\right)\right)\\
&=\frac{1}{T}\operatorname{Tr}\left[A\left(y^0\right)\ldots A\left(y^T\right)\right],
\end{aligned}\qquad\text{(G.3)}
$$

with a single $d\times d$ block matrix

$$
A(y):=\begin{bmatrix}
0 & A_1(y) & 0 & \cdots & 0\\
\vdots & & \ddots & A_2(y) & & \vdots\\
\vdots & & & & \ddots & 0\\
0 & & & & \ddots & A_T(y)\\
A_0(y) & 0 & \cdots & & 0
\end{bmatrix}.\qquad\text{(G.4)}
$$

Its bond dimension is

$$
d=\begin{cases}
r^2+r^4+\cdots+r^T+r^T+\cdots+1, & \text{for even }T,\\
r^2+r^4+\cdots+r^{T-1}+r^{T+1}+r^{T-1}+\cdots+1, & \text{for odd }T,
\end{cases}\qquad\text{(G.5)}
$$

such that $d\le 2r^{T+1}$. Equation (G.3) follows because

$$
\prod_{t=0}^{T}A(y^t)=\begin{bmatrix}
A_0(y^0)A_1(y^1)\ldots A_T(y^T) & 0 & \cdots\cdots & 0\\
0 & A_1(y^0)\ldots A_T(y^{T-1})A_0(y^T) & 0 & \vdots\\
\vdots & 0 & \ddots & \\
0 & \vdots & \ddots\ddots & 0\\
0 & 0 & \cdots\ 0 & A_T(y^0)A_0(y^1)\ldots A_{T-1}(y^T)
\end{bmatrix}.\qquad\text{(G.6)}
$$

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
