# Peer review of "Nonequilibrium steady-state dynamics of Markov processes on graphs"

_SciPost Physics, doi:SciPost Phys. 19, 045 (2025)_

## Round 1 · Referee Report · Anonymous (Referee 1) · 2025-4-21

Strengths

  1. This paper proposes eternal dynamical cavity method to resolve the challenge of describing the non-equilibrium steady state dynamics for a broad range of dynamics models, in particular on sparse random graphs.
  2. The topic is very relevant in practical applications, e.g., disease spreading.
  3. The paper is well-written, giving a very nice introduction of the motivation, sufficient technical details to understand the relevant concepts.

Weaknesses

  1. The continuous dynamics part is very concise. I wonder if the method can be applied to a continuous asymmetric spin model? Could the authors give a concrete experiment?
  2. In Figure 1, the one-time-delayed correlation is compared. But the equilibrium model can be captured by a Boltzmann distribution. I wonder how the one-time-delayed correlation can be computed by a static cavity method. Could the authors offer more details about this?

Report

The paper is well-written, giving a very nice introduction of the motivation, sufficient technical details to understand the relevant concepts. The paper can be accepted after a minor revision.

Requested changes

**Below Eq. 5, A is explained as a matrix, depending on i-th and j-th spin values, which confused me for its explicit form.
**In Eq. 7, the symbol may be a tensor product, please specify.
**In Figure 1, the one-time-delayed correlation is compared. But the equilibrium model can be captured by a Boltzmann distribution. I wonder how the one-time-delayed correlation can be computed by a static cavity method. Could the authors offer more details about this?
***Below Eq. 13, the threshold may be 0.055, as checked from Fig.4.
***The continuous dynamics part is very concise. I wonder if the method can be applied to a continuous asymmetric spin model? Could the authors give a concrete experiment?

Recommendation

Ask for minor revision

---

## Round 1 · Referee Report · Anonymous (Referee 2) · 2025-5-4

Strengths

1- Combines cavity theory with matrix product representation for inference about steady states 2- Good range of test cases both with and without detailed balance (and absorbing states) 3- Method with potential wide applicability

Weaknesses

1- Derivations in the main text are somewhat terse 2- Discussion of continuous time limit (important for many applications) is particularly short

Report

This paper tackles the problem of predicting steady state properties of Markov chains with graphical structure, by using cavity theory and representing trajectory probabilities and messages formally as infinite matrix products. A number of examples are shown where convergence to known baseline results occurs even for modest matrix dimensions d. Extensions to continuous time dynamics are discussed briefly. I would rate this work as significant progress on an important problem, with broad applicability for systems with discrete states.

Requested changes

1- After eq(10), please make clear why complex conjugation (*) is used to define inner products, given that all messages (as probabilities) are real - is the idea to use the method also for complex measures as in calculation of spectral densities? 2- Please comment (at least briefly in the main text, otherwise in an appendix) how the optimisation in (10) is carried out, e.g. is it done by brute force optimisation over A or is there a more efficient method; similarly please comment on how the actual objective function in (10) is evaluated 3- In discussion of continuous time dynamics, please comment on whether the limit \Delta t -> 0 can be taken in the EDC equations (and if not, why not) 4- Please fix typos: Abstract: we apply? this approach p7 princip_al_ eigenvalues Caption fig 6a refers to an error of 0.005, while text says 10^{-3} After (C3b), w -> W App D: an_ iMP distribution (5)

Recommendation

Publish (easily meets expectations and criteria for this Journal; among top 50%)

---

## Round 2 · Author Response

Response to referees

We thank both referees for their careful reading and constructive criticisms, which helped us to substantially improve the manuscript. We address individual comments below:

Referee 1

  1. Below Eq. 5, A is explained as a matrix, depending on i-th and j-th spin values, which confused me for its explicit form.

    We clarified the relation between the matrix-valued function interpretation and the tensor definition.

  2. In Eq. 7, the symbol may be a tensor product, please specify.

    Yes, exactly, this is the Kronecker product. We clarified in the text.

  3. In Figure 1, the one-time-delayed correlation is compared. But the equilibrium model can be captured by a Boltzmann distribution. I wonder how the one-time-delayed correlation can be computed by a static cavity method. Could the authors offer more details about this?

    Thank you for this remark. At difference with the sequential dynamics, the parallel update dynamics does not converge to the equilibrium of the system with the given couplings and fields. Instead, for symmetric couplings, it converges to the equilibrium measure of a related augmented bipartite system. Equilibrium magnetizations in the original system coincide with steady-state magnetizations of the parallel update, and equilibrium correlations of first neighbors in the original system can be recovered as 1 time delayed correlations of first neighbors in the parallel update. This was explained in footnote [28] which refers to [12]. In reality, the explanation is in the SI of [12], so we specified it in the footnote.

  4. Below Eq. 13, the threshold may be 0.055, as checked from Fig.4.

    Thanks a lot for this remark. Indeed there is a discrepancy due to an error in the axis of the figure, the correct threshold is the one in the text. We corrected the figure.

  5. The continuous dynamics part is very concise. I wonder if the method can be applied to a continuous asymmetric spin model? Could the authors give a concrete experiment?

    The scope of this section was to show how, at difference with the transient analysis of [10-12], being an equation that describes the full distribution of trajectories at the steady state trough a "single time" set of parameters (the tensor A), it applies directly also to $\Delta t << 1$, without suffering from an explosion of the calculations with the number of timesteps going to infinity as $1/\Delta t$. It can indeed be applied to an asymmetric spin model, but we chose to compare with the continuous-time SIS model, which has similar characteristics and is extremely standard. Note also that an analytic $\Delta t\to 0$ limit of the equations is in principle possible and desirable but leads to continuous matrix products which require different optimization methods and involve certain technical challenges. See for instance [r1] below. We added this explanation to the text.

Referee 2

  1. After eq(10), please make clear why complex conjugation (*) is used to define inner products, given that all messages (as probabilities) are real - is the idea to use the method also for complex measures as in calculation of spectral densities?

    The referee is perfectly right, the conjugation is superfluous as the tensors we are using are real. However, we prefer to keep it for consistence with the literature (in particular for the truncation methods). We added a comment on this on the manuscript below Eq. 10.

  2. Please comment (at least briefly in the main text, otherwise in an appendix) how the optimisation in (10) is carried out, e.g. is it done by brute force optimisation over A or is there a more efficient method; similarly please comment on how the actual objective function in (10) is evaluated.

    Optimization of (10) is done using the VUMPS algorithm by solving iteratively the set of equations that guarantee stationarity of the fidelity with respect to the matrix A, which in turn involves the repeated solution of two local eigenvalue problems. We added a small comment on this, but refer to [22,24,25] for details.

  3. In discussion of continuous time dynamics, please comment on whether the limit \Delta t -> 0 can be taken in the EDC equations (and if not, why not)

    This is in principle possible but leads to continuous matrix products which require careful definitions and different optimization methods for truncation that involve several technical challenges. See for instance [r1] below. We added this explanation to the text.

  4. Please fix typos: - Abstract: we apply? this approach - p7 princip_al_ eigenvalues - Caption fig 6a refers to an error of 0.005, while text says 10^{-3} - After (C3b), w -> W - App D: an_ iMP distribution (5)

    We thank the referee for these corrections. The correct threshold is the one in the text, we corrected the axis and caption of Fig 6.

  5. [r1] Haegeman, J., Cirac, J. I., Osborne, T. J., & Verstraete, F. (2013). Calculus of continuous matrix product states. Physical Review B—Condensed Matter and Materials Physics, 88(8), 085118.

---

## Round 2 · List of Changes

• We clarified the relation between the matrix-valued function interpretation and the tensor definition below (5) in page 3.
  • Below Eq. 7, we clarified that the tensor product used in Eq. 7 is Kronecker product.
  • We clarified the relation between equilibrium statistics and parallel update statistics in footnote [28].
  • We corrected the axis of Fig. 4.
  • We added a comment on the continuous dynamics and the challenges involved when taking the continuous time limit explicitly. We added reference [32] ([r1] in the response).
  • Below equation (10), we clarified the use of complex conjugation in (10) and clarified that the optimization of (10) is performed by the iterative VUMPS method, referring to [22,24,25] for details.
  • We corrected the caption of Fig 6.

---

## Editorial Decision

published